# FROM LOCAL EXPLAINABILITY TO GLOBAL ROBUSTNESS: IMPROVING THE ROBUSTNESS OF MACHINE LEARNING MODELS USING COUNTERFACTUAL EXPLANATIONS

## ABSTRACT

Sophisticated new adversarial attacks are being introduced at a rapid rate. Such threats have been accompanied by the creation of a wide variety of defense techniques, including robustness techniques. This paper proposes a novel attack-agnostic robustness method that utilizes the local explainability capabilities of counterfactual explanations (CFE data) to improve the robustness of classical machine learning models trained on structured (tabular) data. In order to defend target models, we induce an auxiliary denoising autoencoder (DAE) with benign and CFE data. The DAE serves as a defense mechanism by denoising the input, which can be benign or adversarial, and reconstructing it into the benign data manifold before it is introduced to the target model. We also suggest four protection mechanisms that utilize our DAE, one of which serves as a preventative approach and does not require any changes to the target model. In the other three protection mechanisms, the target model is induced with benign and CFE data in order to both accurately fit the decision boundaries to various samples and improve the model's robustness to diverse perturbations. In our evaluation on three structured datasets, the proposed robustness method achieved results comparable to state-of-the-art robustness techniques which are not attack-agnostic.

## 1 INTRODUCTION

First introduced by Szegedy et al. (2013), adversarial examples are machine learning (ML) model input samples,which are crafted by applying small intentional perturbations that influence the model's classification capabilities. With the discovery of such examples, ML models have proven to be vulnerable to adversaries' ill intent (Chakraborty et al., 2021). Although most adversarial attacks are designed to attack homogeneous data (Chakraborty et al., 2021), recently, new attacks have emerged specifically designed to address the complexities of tabular data (Grolman et al., 2022). As the emerging attacks are more sophisticated than their predecessors, the development of attack-agnostic robustness methods is more critical then ever before.

In the last few years, the need for explainability with regard to ML models and artificial intelligence (AI) methods has become increasingly important. Explainable AI (XAI) allows users and ML developers to better understand how ML models arrive at their decisions (Arrieta et al., 2020). Typically, model-agnostic XAI techniques, which can be adapted to explain any kind of model, are preferable, since they explain the models' decisions without the need to examine the inner technicalities of a given model. One such technique is the "counterfactual explanations" (CFEs) technique, which is a local explainability technique that indicates what adjustments need to be made to the input data based on the ML decision boundaries to enable a user to obtain their desired output. For example, as shown in Figure 1, if P was denied a loan request, a counterfactual example would enable P to understand that if he/she could increase his/her savings by $5,000, the loan request would be approved. This XAI technique is considered local, since it focuses on explaining the behavior of a single sample as opposed to the behavior of the ML model.

In this paper, the local explainability capabilities of counterfactual explanations are utilized in a novel attack-agnostic robustness method effective for heterogeneous tabular data. The method con-

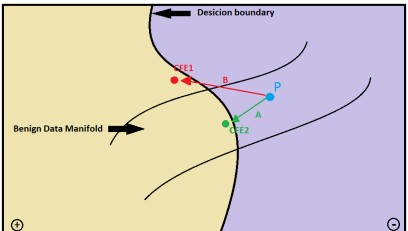

Figure 1: An example for a European online loan application scenario, which adheres to the GDPR. If a person P enters their data as part of a loan application, and the system declines their loan request, the system must provide a counterfactual explanation (CFE) explaining its decision. The endpoints of both the paths (shown in red and green) are valid counterfactual explanations for the original point. While CFE2 indicates that if P had $5,000 more savings the loan would be approved, CFE1 indicates a sufficient scenario but not a necessary one, since it leaves the benign data manifold. The image was inspired by Verma et al. (2020).

sists of the following three steps: *i)* Given benign data, the counterfactual examples (CFE data) are extracted based on the target model's decision boundaries. *ii)* After the extraction of the CFE data, a denoising autoencoder (DAE) is induced with the extracted CFE data. The DAE is trained to reconstruct the noisy input into clean denoised output. *iii)* The extracted CFE data and induced DAE serve as the basis of the four defense mechanisms proposed in this paper.

Unlike most state-of-the-art defense mechanisms that utilize existing adversarial attacks to achieve robustness (Bai et al., 2021), we use CFEs. The greatest disadvantages of using adversarial attacks in defense mechanisms are the ever increasing and evolving variations of attacks and the attack example generation time (Shafahi et al., 2019). For example, for the ART framework (Nicolae et al., 2018), which is one of the most widely used robustness frameworks that generate adversarial examples, there are currently 40 different evasion attacks available. Suppose that there is a need to induce a robust CNN on the MNIST dataset. In this case, training a defense mechanism, such as an attack-based DAE (Sahay et al., 2019), would first require the creation of an adversarially perturbed dataset. To address each known attack and properly defend against it, there would be a need to generate at least 10,000 adversarial examples, for a total of 400,000 adversarial examples for the 40 attacks. With the release of a new attack or a new variation of an existing attack, this process would need to be repeated, making such defenses impractical in real-life scenarios.

In addition, the proposed method has an explainability advantage over attack-based defense mechanisms. CFEs fulfill the requirements of the European General Data Protection Regulation (GDPR) which requires that users receive an explanation in cases in which they are subject to the fully automated decision-making of an algorithm (Voigt & Von dem Bussche, 2017). Furthermore, Wachter et al. (2017) argue that CFEs have three intuitive functions: to increase understanding, provide guidance for future actions, and provide the ability to contest decisions. Finally, when considering the typical user and their ability to make use of the explanations associated with an ML model or AI method, including those used in defense mechanism robustness approaches, the concept of a counterfactual example is much easier to grasp than that of an adversarial example; similarly, understanding how each attack works and the differences between various adversarial attacks, etc. – knowledge which usually takes domain experts years to fully understand – is well out of the scope of the average user.

The main contributions of this paper can be summarized as follows:

- Although XAI techniques have been employed to depict detection methods (Fidel et al., 2020), to the best of our knowledge, we are the first to introduce a method that utilizes XAI techniques to achieve robustness.

- Although local explanations tackle explainability by segmenting the solution space, we demonstrate how they can be generalized to the entire model, providing guidelines on how local explanations can be applied to achieve global robustness.

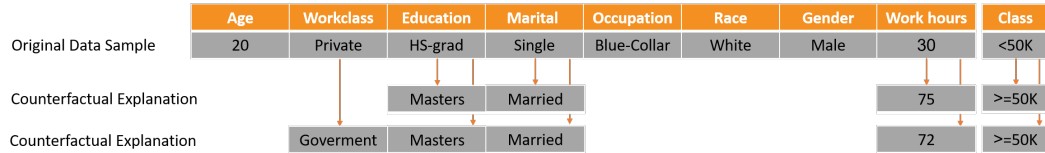

| | Age | Workclass | Education | Marital | Occupation | Race | Gender | Work hours | Class |
|---|---|---|---|---|---|---|---|---|---|
| Original Data Sample | 20 | Private | HS-grad | Single | Blue-Collar | White | Male | 30 | <50K |
| Counterfactual Explanation | | | Masters | Married | | | | 75 | >=50K |
| Counterfactual Explanation | | Goverment | Masters | Married | | | | 72 | >=50K |

Figure 2: An example of counterfactual generation on the Adult Census Data displaying two plausible counterfactual explanations which crossed the decision boundary from not being able to make 50k a year to making more than 50k a year. These explanations could guide a user in his future actions.

- We present an alternative attack-agnostic retraining method, which outlines and reinforces the decision boundaries of the target model. In this method, local counterfactual explanations and their respective labels are combined with the benign training data.

## 2 BACKGROUND

### 2.1 COUNTERMEASURES FOR ADVERSARIAL EXAMPLES

Countermeasure techniques can be categorized into three main groups (Chakraborty et al., 2021; Xu et al., 2020; Yuan et al., 2019): robustness, prevention and detection. In this paper, we will focus mostly on robustness techniques which consist of four main categories (Chakraborty et al., 2021): *(i)* Gradient Masking or Obfuscation which denies the attacker access to useful gradients (applicable only to differential learning algorithms). *(ii)* Provable Defences are methods which are based on theoretical and mathematical guarantees for their success. *(iii)* Regularization methods which are based on controlling and adjusting the weights of the target models and making slight modification to the learning algorithm such that the model generalizes better. *(iv)* Adversarial (re)Training, in which adversarial examples are generated and then the model is explicitly (re)Trained using them. Our suggested method is based on a combination of the first and the last categories. Note, that according to the "No Free Lunch" theorem, a trade-off exist between accuracy and robustness. Therefore, it can not be expected that a model which is enhanced by a robustness method will yield better results (such as higher accuracy) compared to a model which was not.

### 2.2 XAI AND COUNTERFACTUALS

Counterfactual explanations (CFES) in the context of ML were first introduced by Wachter et al. (2017) in 2017 and are an emerging technique of local explainability. CFES explain a prediction by calculating a change (usually minimal) in a given data point that would cause the underlying ML model to classify it in the desired class. The interpretability of CFES forms an effective tool for the average user. In addition, data scientist and engineers can leverage CFES to adjust the ML model accordingly.

Many XAI frameworks allow the generation of CFES. For example, Dice-ML creates CFES for binary classification problems taking into account the constraints and complexity of tabular data (Mothilal et al., 2020). Alibi (Klaise et al., 2021), on the other hand, being a general XAI frameworks includes many explainability techniques such as CFES, however, it does not take into account the complexity of tabular data during CFES generation.

Since CFES and adversarial example are both obtained by solving the same optimization problem, the research community extensively explores the similarity and differences between them.

$$\arg\min_{x \in X} d(x, x') + \lambda d'(f(x'), y_{des}) \tag{1}$$

While CFES originated from the philosophy field, adversarial examples have strong origins in the reliability and robustness literature in computer science (Freiesleben, 2022). Grath et al. (2018) claimed that CFES and adversarial examples are similar in a sense that both are example-based approaches. However, the distinction between CFES and adversarial examples is described as the

difference between flipping and explaining decisions. CFES inform about the changes, while adversarial examples aim at hiding those. Freiesleben (2022) presented two definitional differences that have so far been overseen: *(i)*Misclassification is necessary for an adversarial example otherwise it provides no means to attack, while CFES are agnostic since a correctly classified CFE is not only acceptable but sometimes desirable. *(ii)*Proximity to the original input is significant for CFES, because without maximal closeness a CFE can show a sufficient scenario but not a necessary one. On the other hand adversarial examples can benefit both from close proximity (less perceptible) and far proximity (transferability). Kenny & Keane (2021) reinforced Freiesleben (Freiesleben, 2022) research, by introducing the "semi-factual explanation" term (correctly classified CFE) into the context of ML and demonstrating the significance of proximity to the original input both in CFE and semifactual explanations.

## 3 RELATED WORK

### 3.1 MAG-NET

Mag-Net (Meng & Chen, 2017), a framework for defending against adversarial examples, consists of two main components, a detector autoencoder and a reformer autoencoder, both of which are trained on benign samples. Upon receiving an input $x$, the detector autoencoder uses the reconstruction error $E$ of the samples. If $E(x)$ is above a certain predefined threshold, the input $x$ is rejected. Otherwise, it is passed on to the reformer, which attempts to reform $x$; if it is a benign sample, its classification should not change, and if it is an adversarial sample, the change should be sufficient to ensure that the reformed sample $x'$ is close to the benign data manifold. Once the sample $x$ is reformed to $x'$ and all of its adversarial features have been denoised, it is passed on to the classification model for the final prediction.

### 3.2 CASCADED AUTOENCODER APPROACH

Sahay et al. (2019) proposed the use of a DAE to defend against the fast gradient sign method (FGSM) attack. First, a DAE that was trained on both FGSM and benign data is used to denoise the data. Then, the dimension of the denoised output is reduced, using a hidden layer representation of another autoencoder, and passed on to the target model which was trained on data with the respective dimension. Their evaluation showed that the performance of the denoising portion of the cascaded defense was comparable to the performance obtained when the output underwent dimensionality reduction. A similar state-of-the-art method was recently proposed by Ryu & Choi (2022).

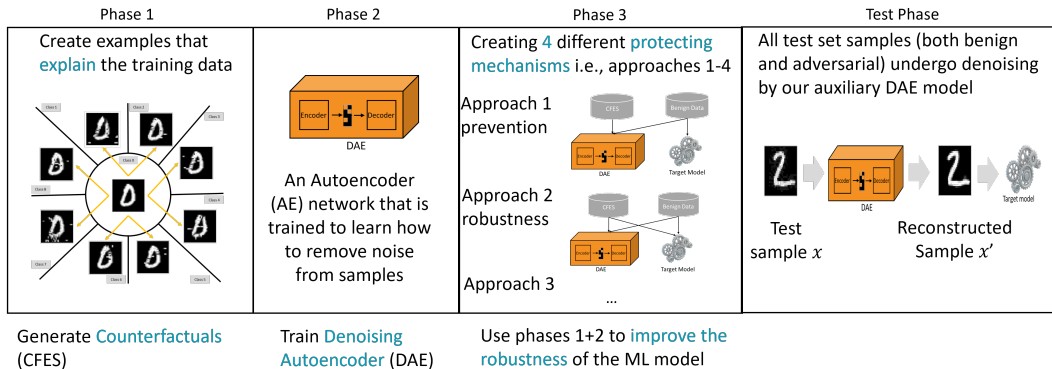

Figure 3: The proposed method.

## 4 PROPOSED METHOD

The proposed method uses the denoising features of a DAE to bring adversarially perturbed samples back into the benign data manifold before being introduced to the model, acting as an "AI Firewall." The method consists of three main phases, as shown in Figure 3 :

- Generate counterfactual examples (CFEs).
- Train a denoising autoencoder (DAE).
- Use the created CFEs and the DAE to improve the robustness of the ML model.

## 4.1 GENERATING COUNTERFACTUAL EXPLANATIONS

To generate CFEs, we used the DiCE ML (Mothilal et al., 2020) framework, which, in order to better interpret the decision boundaries of the model, requires both the data used to train the target model and the target model itself. When considering binary classification problems, DiCE ML (Mothilal et al., 2020) allows the creation of two types of CFEs: (1) Cross-class projecting, CFEs from class "0" to class "1" or vise versa. In a robustness against perturbations defense method such as ours, the projecting direction depends on the adversary's motivation with respect to the examined dataset. (2) Same-class projecting, CFEs which are also called semifactuals (Kenny & Keane, 2021). In the first phase, the DiCE ML framework was applied on each benign dataset, generating a dataset consisting of cross-class projecting counterfactual explanations.

## 4.2 TRAINING THE DAE

In the second phase, after generating sufficient CFEs for a single class or for each of the classes (depending on the attacker's motivation with respect to the examined dataset), the original dataset (benign samples) is merged with the CFE dataset (created in the previous phase). The merged dataset is referred to as the "From Dataset" for the DAE. For each CFE $x'$ in the CFES dataset, we aggregate the original sample $x$ (from which $x'$ was created) into a new dataset of samples. This dataset is merged with the benign dataset, creating the "To Dataset." Then the DAE is trained to reconstruct the "From Dataset" into the "To Dataset."

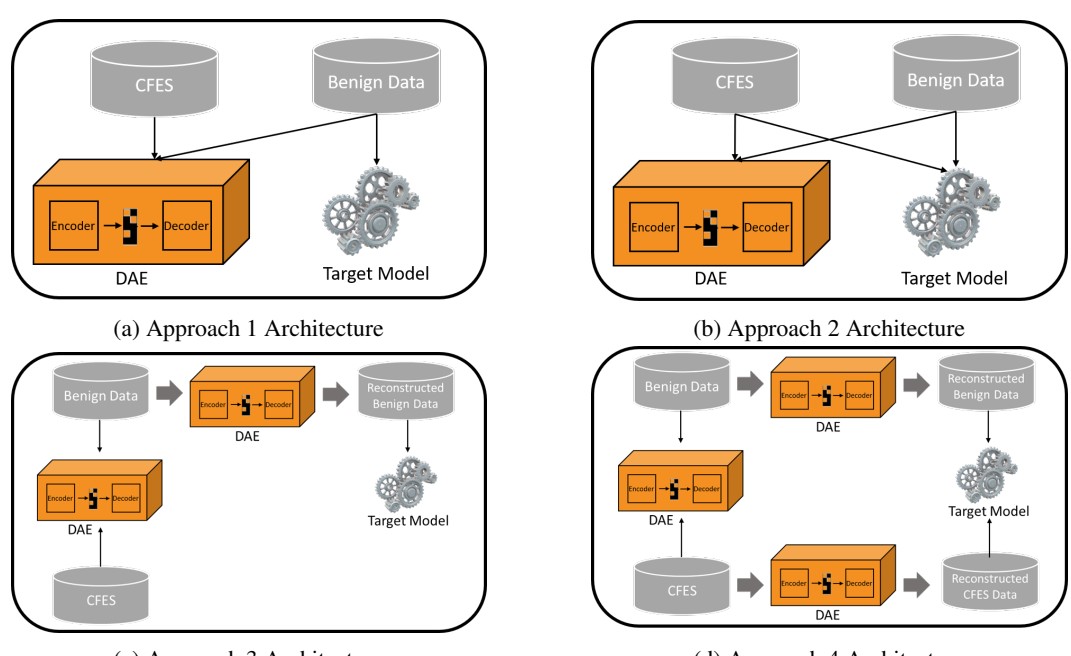

(a) Approach 1 Architecture     (b) Approach 2 Architecture

(c) Approach 3 Architecture     (d) Approach 4 Architecture

Figure 4: The four approaches used in the proposed method. The thin line denotes the data used to train each model, while the thick line denotes the application of a model on the specified data.

## 4.3 USING THE CFES AND THE DAE

After having trained the DAE and generated the CFES, in the third phase, they can be used in a variety of different approaches, as presented in Figure 4:

- Approach 1: The DAE serves as an AI firewall,located before an untouched target model (prevention), shown in Figure 4a.

- Approach 2: The DAE serves as an AI firewall, located before the target model which was trained on the created CFEs as well as the benign samples (robustness), shown in Figure 4b.

- Approach 3: The DAE serves as an AI firewall, located before the target model which was trained on the reconstructed benign samples produced by the DAE (robustness), shown in Figure 4c.

- Approach 4: The DAE serves as an AI firewall, located before the target model which was trained on the reconstructed benign samples and the reconstructed CFE samples, both of which were produced by the DAE (robustness), shown in Figure 4d.

### 4.4 INFERENCE TIME

During the inference (test) phase, both adversarial and benign samples may be encountered. Upon receiving a sample, the DAE model uses its denoising features to reconstruct the sample into a new denoised sample, which will be passed on to the target model for classification. A properly trained DAE model should ensure that the benign samples remain in their respective classes, while denoising the adversarial examples sufficiently to bring them back into the benign data manifold.

## 5 EVALUATION

### 5.1 DATASETS

**The Hateful Users on Twitter** dataset contains a network of $100K$ users, of which, $4,972$ samples are samples from English speaking Twitter users that were manually annotated as hateful or non-hateful users (Ribeiro et al., 2018). Each record contains several content-related, network-related, and activity related features such as "Number of hashtags,""Followers Count," and "Hateful Neighbor," respectively. The dataset includes 1,000 features and just $544$ samples of annotated hateful users, meaning that it is an unbalanced dataset. This dataset was preprocessed as suggested by Grolman et al. (2022), which allows the use of the state-of-the-art HateVersarial attack in the evaluation of our method.

**The original German Credit Risk** contains $1,000$ samples with 20 attributes. Each record represents a person who takes a credit by a bank. Each person can be classified as good or bad credit risk based on their set of attributes. We used a version of this dataset[1] that was already preprocessed and underwent feature selection, which contains the same amount of samples as the original dataset but just the nine most important features. Of the nine features, five are categorical and were encoded based on the remaining features [2], while the other four were normalized using min-max-normalization.

**The Adult/Census Income** dataset contains $48,842$ samples with 14 categorical and numerical features. Each record contains a person attributes. A person can be classified as exceeding an income of $\$50K/yr$ or not based on the census data. From the 14 features in this dataset, we selected the top eight features based on previously performed data analysis[3]. Six of the remaining features are categorical and were one-hot encoded. Nearly $25\%$ of the data belongs to adults that make more than $50K$ a year, making the dataset fairly balanced; therefore, no additional balancing was needed.

**The Phishing** dataset contains 11,000 samples of URLs, each of which contain 30 website parameters and a class label identifying it as a phishing website or not. All features are categorical with a range between $(-1, 1)$.[4]

---

[1]https://www.kaggle.com/datasets/uciml/german-credit
[2]https://towardsdatascience.com/deep-embeddings-for-categorical-variables-cat2vec-b05c8ab63
[3]https://rpubs.com/H_Zhu/235617
[4]https://www.kaggle.com/datasets/eswarchandt/phishing-website-detector

## 5.2 Experimental Settings

All experiments were performed on CentOS Linux 7 (CORE) operating system using 24G of memory and a NVIDIA GeForce RTX 3090 Ti graphics card. The code used in the experiments was written in Python 3.8.12, scikit-learn 1.1.1, NumPy 1.19.5, and TensorFlow-GPU 2.8.0.

All of the tabular datasets were split into three parts, 80% for training, 10% for validation, and 10% for testing. During cross-validation the models were trained on a training set that was created from the original training set by choosing samples with replacement. The process of choosing these samples was seeded with a different seed for each experiment. Since we applied a black-box scenario, the dataset was split into three main parts: (1) Training: training the target model, generating CFEs, and training the autoencoder; (2) Validation: validating the trained models; and (3) Testing: evaluating the target model and generating the attacks.

In terms of time complexity, the training time of the target model and the autoencoder is negligible, but the CFE generation time depends on the size of the dataset and the number of CFEs wanted.

## 5.3 Attack Settings

When evaluating our method, three state-of-the-art attacks were performed, two of which were part of the Adversarial Robustness Toolbox (Nicolae et al., 2018): (1) HopSkipJump attack (HSJ) (Chen et al., 2020),(2) Zeroth Order Optimisation attack (ZOO) (Chen et al., 2017), and an attack designed specifically for tabular data: HateVersarial (Grolman et al., 2022).

All of these attacks were performed in a black-box manner. For adversarial sample generation, HSJ did not require any changes to the default parameters provided by the Adversarial Robustness Toolbox. For the ZOO attack the parameters were set as follows: max_iter=160, nb_parallel=15, learning_rate=5, and variable_h=1. While no hyperparameters were needed for HateVersarial, the constraints of each feature needed to be specified.

## 5.4 Target Model and DAE Auxiliary Model Settings

**The target model** employed was a TensorFlow random forest classifier with no hyperparameter initialization except for the random state. We choose to utilize a random forest classifier, since nondifferential learning algorithms usually obtain better performance for learning tasks with tabular data (Arik & Pfister, 2021; Lundberg & Lee, 2017; Popov et al., 2019), and among those, random forest and XGBoost obtain the best performances. XGBoost was not included in our evaluation, since the DiCE ML framework does not support counterfactual generation for the XGBoost model.

**The DAE Auxiliary Model** used in the evaluations was a variational autoencoder (VAE), since by design such autoencoders are more robust to adversarial perturbations than their deterministic counterparts (Camuto et al., 2021). The VAE consists of two hidden linear layers in the encoder and decoder, with H1 and H2 representing the size of the layers and latent_dim representing the size of the latent dimension. The VAE employs the KL divergence loss function and utilizes the reparameterization trick; between each layer, 1D batch normalization is performed.

## 5.5 Performance Comparison

We evaluated the performance of our four protection mechanisms and compared the results to the results of the following state-of-the-art methods: **Benign DAE**: This DAE is based on the reformer DAE (Meng & Chen, 2017). **Attack-based DAE**: This DAE is trained on both benign and adversarially perturbed data (adversarial re/training for DAEs) (Sahay et al., 2019).

## 5.6 Results

Table 1 presents the results obtained by each of the evaluated methods on each dataset. The attack column indicates which of the three attacks were employed on the test set. The target model column presents the accuracy of the untouched target model on the test set. It also indicates the model's reduced accuracy resulting from each of the adversarial attacks. The DAE columns present the accuracy of the target model using the benign DAE and attacked based DAEs (described in the

| | | Mean Target Model Classification Accuracy with standard deviation | | | | | | | |
|---|---|---|---|---|---|---|---|---|---|
| Data | Attack | TM | BDAE | HDAE | ZDAE | A1 | A2 | A3 | A4 |
| Twitter | Benign | 0.9429 | **0.8608** | 0.8576 | 0.8563 | 0.8589 | 0.8595 | 0.8503 | 0.8451 |
| | | ±0.0081 | **±0.0114** | ±0.0155 | ±0.0147 | ±0.0177 | ±0.0139 | ±0.0259 | ±0.0285 |
| | HSJ | 0.702 | 0.8594 | 0.873 | 0.8717 | **0.8724** | **0.8724** | 0.8601 | 0.8581 |
| | | ±0.0223 | ±0.0069 | ±0.0 | ±0.0022 | **±0.0028** | **±0.0011** | ±0.0091 | ±0.0165 |
| | HV | 0.0439 | 0.7129 | 0.7362 | 0.7362 | 0.7417 | **0.7445** | 0.7307 | 0.717 |
| | | ±0.0223 | ±0.0157 | ±0.0 | ±0.0 | ±0.0054 | **±0.0091** | ±0.0248 | ±0.0234 |
| | ZOO | 0.6062 | 0.8601 | 0.873 | **0.8724** | 0.8555 | 0.8529 | 0.8296 | 0.7979 |
| | | ±0.021 | ±0.0111 | ±0.0 | **±0.0011** | ±0.0211 | ±0.022 | ±0.0223 | ±0.0231 |
| Adult | Benign | 0.8167 | 0.7081 | 0.6501 | 0.662 | 0.645 | **0.7642** | 0.754 | 0.7601 |
| | | ±0.0021 | ±0.027 | ±0.0143 | ±0.0102 | ±0.0041 | **±0.022** | ±0.0013 | ±0.0033 |
| | HSJ | 0.8602 | 0.7406 | 0.5955 | 0.6819 | 0.6641 | 0.7517 | **0.755** | 0.7489 |
| | | ±0.0019 | ±0.0169 | ±0.044 | ±0.0127 | ±0.0077 | ±0.0197 | **±0.0046** | ±0.0096 |
| | HV | 0.1762 | 0.5132 | 0.529 | **0.5349** | 0.5034 | 0.498 | 0.4938 | 0.5022 |
| | | ±0.0037 | ±0.0056 | ±0.0104 | **±0.0036** | ±0.0161 | ±0.0058 | ±0.0062 | ±0.0086 |
| | ZOO | 0.4574 | 0.6778 | 0.6407 | 0.6218 | 0.6029 | **0.7436** | 0.7174 | 0.7285 |
| | | ±0.022 | ±0.0135 | ±0.0374 | ±0.0219 | ±0.0343 | **±0.0133** | ±0.0204 | ±0.0167 |
| German | Benign | 0.7 | 0.6875 | 0.695 | 0.7 | 0.675 | **0.7125** | 0.6125 | 0.615 |
| | | ±0.0324 | ±0.0335 | ±0.032 | ±0.0332 | ±0.045 | **±0.0286** | ±0.1035 | ±0.0757 |
| | HSJ | 0.6275 | **0.625** | 0.595 | 0.5875 | **0.625** | 0.6025 | 0.545 | 0.5775 |
| | | ±0.0228 | **±0.025** | ±0.005 | ±0.0192 | **±0.013** | ±0.0669 | | ±0.0408 |
| | HV | 0.39375 | 0.48125 | 0.5375 | 0.525 | 0.5125 | **0.54375** | 0.46875 | 0.53125 |
| | | ±0.037 | ±0.067 | ±0.0279 | ±0.0306 | ±0.0375 | **±0.0108** | ±0.0325 | ±0.0325 |
| | ZOO | 0.3875 | 0.5575 | 0.5625 | 0.585 | 0.5525 | **0.61** | 0.525 | 0.5425 |
| | | ±0.037 | ±0.0427 | ±0.0593 | ±0.0218 | ±0.0614 | **±0.0** | ±0.0335 | ±0.0179 |
| Phishing | Benign | 0.9778 | **0.8433** | 0.4457 | 0.6767 | 0.8214 | 0.7448 | 0.8096 | 0.811 |
| | | ±0.0075 | **±0.0203** | ±0.0 | ±0.0184 | ±0.012 | ±0.0366 | ±0.0115 | ±0.0154 |
| | HSJ | 0.2732 | **0.7929** | 0.4557 | 0.6331 | 0.7649 | 0.707 | 0.7631 | 0.764 |
| | | ±0.0384 | **±0.0081** | ±0.0 | ±0.0233 | ±0.0208 | ±0.009 | | ±0.0203 |
| | HV | 0.0342 | 0.4693 | 0.4551 | 0.4475 | 0.6359 | 0.6285 | 0.635 | **0.6394** |
| | | ±0.0034 | ±0.0137 | ±0.0 | ±0.0112 | ±0.0154 | ±0.0171 | ±0.0167 | **±0.018** |
| | ZOO | 0.0929 | 0.2271 | 0.4557 | 0.467 | 0.4186 | **0.4749** | 0.429 | 0.4459 |
| | | ±0.0101 | ±0.0196 | ±0.0 | ±0.0147 | ±0.0199 | **±0.0204** | ±0.0346 | ±0.0214 |

Table 1: Evaluation results on tabular datasets (in the table, TM denotes target model; BDAE, HDAE, and ZDAE denote denoising autoencoders trained on benign, HopSkipJump, and ZOO data; A1-A4 denote our approaches).

Performance Comparison section above). Note that there is no DAE HateVersarial column, since HateVersarial is an iterative attack method based on feature importance and generating a full adversarial training set is not possible given the time required. The columns A1-4 present the accuracy obtained by our proposed protection mechanisms, with their respective settings.

### 5.6.1 ATTACKS' IMPACT ON THE TARGET MODEL

It can be seen that on most datasets, the HateVersarial attack has the highest reduction in accuracy, since it is a nonuniform perturbation attack and was specifically designed to account for the complexities of tabular data. The ZOO attack achieved slightly lower accuracy reduction on the Adult and Twitter datasets but outperformed the HateVersarial attack on the German Credit dataset. The HSJ attack has the lowest overall performance across all of the datasets except from the Phishing dataset.

### 5.6.2 DAE RESULTS

As can be seen in the DAE columns, the benign DAE (Meng & Chen, 2017) trained only on benign data already provides a relative boost in robustness. However, the baseline DAE is outperformed by the adversarially trained DAEs (Sahay et al., 2019). In examining both the DAE columns and the A1-4 columns, it can be seen that our proposed approaches have relatively similar results to the attack-based DAEs but mostly outperform them. However, it is important to emphasize that our method is an attack-agnostic method and requires no previous knowledge about the attacks. Moreover, the attack-based DAEs rely only on existing attacks and would need to be retrained in order to defend against the release of new variations of an existing attack or new attacks. This can be seen particularly in the case of the HSJ DAE on the Phishing dataset which has a a hard time generalizing across different attacks.

### 5.6.3 EXAMINING THE DIFFERENT APPROACHES

In examining the four proposed approaches, it can be seen that the third and fourth approaches obtain the most stable results across all of the datasets, but both require retraining of the target model and the generation of reconstructed samples which can be quite time consuming and inconvenient for large models and/or datasets. Comparing the first and second approaches, it can be seen that the second approach is more stable across the datasets in terms of robustness, but it requires a one-time retraining of the target model which can be inconvenient for large models. The first approach is not as stable as the second approach, but it does not require any retraining at all and performs quite well compared to the attack-based DAEs. Therefore, the use of the third and fourth approaches is recommended in cases in which the user has a lot of time or are deploying the ML model for the first time. If time is of the essence, then the second approach is recommended over the third or fourth approach. If the user is not interested in retraining the model and only has time to generate CFEs, then the first approach should be employed.

### 5.7 DISCUSSION

When evaluating our results, one could argue that it is sufficient to re/train the denoising autoencoder on a single specific attack in order to provide an adequate defense against other attacks. However, when examining the Phishing dataset results, one cannot ignore the fact that the HopSkipJump denoising autoencoder obtained the lowest results. In addition, it has been shown that adversarial training defenses tend to not generalize across different attack strategies, thus leaving the target model vulnerable to new/unknown attack variations (Bakhti et al., 2019). Furthermore, the experiments of Sahay et al. (2019) found that a DAE trained on a $l_2$ perturbed dataset performs worse than a DAE trained on a $l_\infty$ perturbed dataset even when combating $l_2$-based attacks.

## 6 CONCLUSION AND FUTURE WORKS

As a defense against adversarial attacks, in this paper, we presented an attack-agnostic robustness method which uses counterfactual explanations. Generally attack-agnostic methods are preferable as a robustness measure, because they can combat the never-ending releases of new attack methods. Our evaluation on three structured datasets showed that the proposed robustness method, when used in four new protective mechanisms, achieved results comparable to state-of-the-art robustness techniques which do not have the advantage of being attack-agnostic. Future work could include extending our method for unstructured data such as images, where when creating CFEs and considering multi-label classification problems (such as MNIST), given a sample from a specific class, it is advised to project it onto every other class using CFE generation. This process should be repeated for several training samples from all the class types to generate a dataset of CFE samples. This approach can be applied on unstructured data but is not limited to it such as for multi-class structured data problems.

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
