# OpenReview forum: "From Local Explainability to Global Robustness: Improving the Robustness of Machine Learning Models Using Counterfactual Explanations"
_ICLR.cc/2024/Conference — ICLR 2024 Conference Withdrawn Submission_

### Official Review · Reviewer_ikyJ · 2023-10-30

**Soundness:** 2 fair
**Presentation:** 3 good
**Contribution:** 1 poor
**Rating:** 3
**Confidence:** 4

**Summary:**

The paper examines adversarial defense through model explanations. The method suggests using counterfactual examples to train a denoising autoencoder. This autoencoder reconstructs benign samples for training target models. The target model is said to be resistant to black-box attacks and correctly classifies adversarial examples. Experiments on four structured datasets demonstrate that the proposed method achieves similar results to other baseline models.

**Strengths:**

The paper is well-written and easily understandable.

**Weaknesses:**

1. The proposed method has limited application. It has only been validated on a binary structured dataset. Additionally, based on the description, the defense setting appears relatively easy since the defender has full access to the target model while the attackers do not.
2. The experimental results are unconvincing. Table 1 shows that nearly 40% of the tested accuracies are lower than the baselines. The transferability of the proposed method can also be examined. For instance, can the DAE trained on one dataset be applied to other datasets? It would also be valuable to present those results.
3. [1] has demonstrated that the BPDA can break the denoising operation on input samples. However, the paper lacks any discussion about the effects of BPDA on the proposed method. The experiments should include similar results evaluated on those attacks.
4. The paper solely proposes a defense framework without conducting a rigorous study on the reasons behind the proposed method.
5. The claim of "a total of 400,000 adversarial examples for the 40 attacks" on Page 2 has not been verified. For instance, the certified defenses can correctly classify all adversarial examples within the attack budgets regardless of the attack types.

[1] Obfuscated Gradients Give a False Sense of Security: Circumventing Defenses to Adversarial Examples

**Questions:**

See weaknesses

---

### Official Review · Reviewer_8JLu · 2023-10-31

**Soundness:** 2 fair
**Presentation:** 4 excellent
**Contribution:** 3 good
**Rating:** 5
**Confidence:** 4

**Summary:**

The primary focus of this paper lies in leveraging Counterfactual Examples (CFEs) to enhance the robustness of machine learning models against adversarial attacks in the context of tabular data. The authors propose to use CFEs since they are independent of the large variety of attack types. To defend the models against adversarial attacks, the authors propose the incorporation of a Denoising Autoencoder (DAE) trained using benign data as well as CFEs. This DAE is deployed during testing, where it serves as a firewall by denoising adversarial examples, aiming to bring them back to the benign class, therefore making them less harmful to the model's performance. The model protected by the DAE can either be trained on i) benign examples, ii) a mix of CFEs and clean examples, iii) denoised benign examples, iv) denoised benign and CFE samples.

**Strengths:**

- Originality: The paper introduces a novel approach to enhancing the robustness against adversarial attacks using Denoising Autoencoders (DAEs) and counterfactual examples (CFE). The integration of DAEs and CFEs represents a fresh perspective, demonstrating originality in the paper's contribution.

- Clarity: The paper is entirely clear, and it presents the approach in a well-explained manner that allows to easily replicate it even without access to the original code

- Quality: The paper is qualitative regarding the rigorous proposed approach, evaluation on relevant datasets for tabular data and comparison with a range of baselines.

- Significance: The paper adds value to the important field of adversarial attack mitigation in machine learning. By demonstrating that CFEs can enhance the robustness of models against adversarial attacks, the paper contributes to the development of more secure and reliable machine learning systems. Considering the high degree of explainability for CFEs, the approach is important for having more transparent and easily understandable robustness mechanisms.

**Weaknesses:**

The primary weakness of this paper lies in the absence of a guarantee and concrete evidence to support the claim that CFEs (Counterfactual Explanations) robustification, despite its attack-agnostic nature, can effectively defend against a wide range of attacks, in contrast to single attack robustification.

- Firstly, the paper acknowledges that adversarial defenses typically struggle to generalize across different attacks. However, the experimental evaluation across four datasets and two defense mechanisms reveals only one instance where one of the defenses did not exhibit strong generalization.

- Secondly, when the authors compare attack-based DAE (Denoising Autoencoder) defenses with CFE-based defenses, the comparison settings are not entirely equivalent. In three out of the four CFE-based defenses (scenarios A2, A3, and A4), denoised examples are employed during the training phase, a practice not used in adversarial-based defenses. This raises the question of what the results would be if adversarial examples were used in scenarios A2, A3, and A4 instead of CFEs. The only scenarios that would enable a somewhat equivalent comparison involve the use of denoised examples at test time such as BDAE, HDAE, ZDAE, versus A1. In such case, CFE-based defenses often do not outperform the ones using adversarial examples.

- Lastly, the techniques for generating CFE examples vary, and in some instances, they resemble to the attacks they are designed to defend against (i.e genetic algorithm approach). This prompts a discussion on whether certain CFE-based defenses may perform better against specific types of attacks.

A second notable weakness is the paper's failure to account for adaptive attacks. The core element of the defense proposed in this paper is the DAE, and if attackers are aware of this, they could fine-tune their examples to induce high loss in the DAE. This is a well-explored scenario for DAE defenses in computer vision.

**Questions:**

Did you try using adversarial examples instead of CFEs for any of the scenarios A1-A4?

---

### Official Review · Reviewer_GewD · 2023-10-31

**Soundness:** 2 fair
**Presentation:** 2 fair
**Contribution:** 3 good
**Rating:** 5
**Confidence:** 4

**Summary:**

This paper proposes a preprocessing defense based on noise removal against adversarial attacks. The method leverages counterfactual explanations (CFE) to enhance the robustness of classical machine learning models on tabular data. An auxiliary denoising autoencoder (DAE) is employed, which uses benign and CFE data to denoise inputs. Four defense strategies using this DAE are proposed: the first one does not alter the target model, while the other three refine the target model using benign and CFE data for improved accuracy and robustness. Experiments are performed on four tabular datasets.

**Strengths:**

# Novelty

- I am not aware of any prior work using CFEs to defend against adversarial examples. It is an interesting idea to connect explainability to adversarial defenses.

**Weaknesses:**

# Prior work

- The novelty of the paper resides in using CFEs as data for training the denosing autoencoder. Beyond that, the method does the same as most other denoising and preprocessing defenses in literature. However, the representation of prior work in the paper is done based on only two references. I suggest including more work from this field where publications in the same direction as current work are abundant.

# Soundness

- Assumptions: no threat model is presented for the paper. It becomes clear after reading the paper that the defense is designed under black-box assumptions for the adversary. However, this is only presented implicitly in the Experiments section.
- Assumptions: it seems DiCE ML, one of the main components of the proposed method, is only capable of binary classification. Moreover, it needs access to both the trained model and the training set. These limitations should be mentioned as part of the proposed method.
- More generally, the limitations of the proposed method are not addressed in the paper, except maybe some of them being mentioned for the first time in the conclusion. It is common practice to not introduce new information in the conclusion. Moreover, limitations deserve a dedicated discussion in the paper.

# Significance

- The scope of applicability of the proposed method is quite limited: black-box attacks on tabular data for classical ML models, with only one type of model being evaluated (random forest classifier). Moreover, the paper states that the method cannot be used with XGBoost, one of the most popular and performant models on tabular data.
- Black-box attacks are usually not as strong as white-box ones, as the attacker does not have access to model internals. Proposing a method that is only capable of defending against weak attackers does not seem particularly impactful.
- Even limited to the black-box setting, one would expect to see more recent and impactful attacks included in the experiments (e.g., [boundary attack](https://arxiv.org/abs/1712.04248), [square attack](https://arxiv.org/pdf/1912.00049.pdf)).
- The experiments oppose the proposed method to versions of DAE for adversarial defense. As such, this looks more like an ablation study than a comparison to other defenses. When the contribution of a paper is a new defense, it makes sense to compare it to state-of-the-art defenses.


# Clarity

- The explanation of the datasets used to train the DAE (Sec. 4.2) is not very clear. Perhaps writing the mathematical notation with the pairs of (input, output) that represent the training samples would help.
- The training of the target model with CFEs also needs to be justified and explained. What exactly would be the training procedure?
- Additional proofreading seems necessary. See some examples below.

# Minor points

- The abstract does not seem to be up to date on how many datasets are used in the experiments: three instead of four.
- samples,which -> samples, which
- (re)Trained -> (re)trained.
- Counterfactual explanations (CFES) -> Counterfactual explanations (CFEs). Sometimes used CFES or CFEs. Please use consistently.
- CFES and adversarial example are both -> CFES and adversarial examples are both
- two definitional differences -> two defining differences
- (i)Misclassification -> (i) Misclassification
- "AI firewall,located" -> "AI firewall, located"

**Questions:**

1. What classical machine learning models can use the proposed defense?
2. Please see comments above.

---

### Official Review · Reviewer_9mE5 · 2023-11-09

**Soundness:** 2 fair
**Presentation:** 2 fair
**Contribution:** 2 fair
**Rating:** 3
**Confidence:** 3

**Summary:**

This papaer introduces a new method to improve model robustness by utilizing the explainability of generated CFE data. After constructing the CFE data, a denoising autoencoder was applied for building reconstruction-based four different defending pipeline. The overall pipeline is flexible and easy to implement. This paper also conduct experiments on four small datasets, and evaluate their proposed defending method against several black-box attacks (HSJ, ZOO, HV). and its proposed method could achieve marginal robustness improvement compared to AE baselines. No further ablation studies on different defending pipelines was made and no white-box robustness evaluation was conducted.

**Strengths:**

- Motivation is interesting: It would be interesting to see how these CFE data (which are generated close to decision boundary) could help further improve the model robustness.
- The whole pipeline is computational efficient compared to other adversarial training or smoothness training pipelines.

**Weaknesses:**

Most of my concerns are placed in question part:

- The overall novelty is quite limited: detection with CFE-data trained AE looks novel but there is no strong point on explaining why CFE-data trained AE could perform better than other denoising AE which are trained on either augmented data or adversarial examples (e.g., MagNet or Defense GAN). It makes me feel like the whole paper is just applying AE on the new set of samples (which are close to the decision boundary for sure) but without too much underlying reason or comparison to other AE baselines.
- Experiment part is not solid: No sufficient baseline included in main result table. Also the datasets are too small - we cannot be convinced by the robustness analysis on toy datasets with 14 or 20 or 30 feature dimensions. More natural image datasets results need to be added.
- Bad writing on methodology part: hard to follow how each approach is actually applied.

**Questions:**

Major concerns:
- Methodology part: Section 4.3 looks super unclear: four different pipelines are described in a very vague way and I feel it is quite hard to follow: what is the "AI firewall" and how it can be well configured (e.g. detection threshold). Though I spent a lot of time understanding what's going on here, I still cannot figure out the differences between approach 1 and 2 - In Section 4.2 you mentioned that both "From dataset" and "To dataset" contain benign samples right? Then I was wondering what is the exact difference between approach 1 and 2?
- Evaluation part: The chosen dataset are too small and not convincing enough. I'm curious about why most of the datasets this paper used are with very few attributes? Back to MagNet, they performed their evaluation on both MNIST and CIFAR datasets - can we have the comparable results on MNIST/CIFAR10 and include the MagNet results as baseline?
- Evaluation part: Considering one of the mentioned related work, MagNet, has been proven to be vulnerable under grey-box threat model setting [1], it would be good to show if the proposed CFE-data motivated method could achieve better robust accuracy under the transferable attack scenario.
- Experimental details: I cannot find any DAE training / CFE-data generation details (model arch, epochs, lr, decay etc.), and these information are very important.


[1] Carlini, Nicholas, and David Wagner. "Magnet and" efficient defenses against adversarial attacks" are not robust to adversarial examples." arXiv preprint arXiv:1711.08478 (2017).